# Prevalence and morphometry of the fabella in cadaveric and radiological studies: Is there a link to knee osteoarthritis?

Ebru Yolaçan[1]*, Nurcan Ercıktı[2], Barış Baykal[3], Uğur Bozlar[4], Necdet Kocabıyık[2], Mehmet Ali Güner[2], Necati Salman[2], Ayşe Özdemir[3], Mislina Utlu[2], Laçin Ramazanlı[5]

1 Department of Anatomy, Samsun University Faculty of Medicine, Samsun, Türkiye, 2 Department of Anatomy, University of Health Sciences Gülhane Faculty of Medicine, Ankara, Türkiye, 3 Department of Histology and Embryology, University of Health Sciences Gülhane Faculty of Medicine, Ankara, Türkiye, 4 Gülhane Faculty of Medicine, Department of Radiology, University of Health Sciences, Ankara, Türkiye, 5 Department of Radiology, Azerbaijan Clinical Medical Center, Baku, Azerbaijan

* yolacanebru@gmail.com

## Abstract

### Objective

To investigate the morphometric features of the fabella in cadaveric specimens and evaluate its prevalence and potential association with knee osteoarthritis through radiological analysis.

### Methods

In the cadaveric study, 65 knees from 32 cadavers and 3 leg specimens were dissected. Fabellas were measured and analyzed histologically. In the radiological study, bilateral knee X-rays of 1712 patients with osteoarthritis and 2304 control subjects were retrospectively reviewed. The presence and laterality of the fabella were recorded.

### Results

Fabella was detected in 18.4% of dissected knees. No significant sex-based differences were found in morphometric measurements (p > 0.05). In the radiological study, the overall prevalence was 36.5% (37.6% in patients, 35.7% in controls), with no statistically significant association between fabella presence and osteoarthritis (p > 0.05). Lateralization patterns did not differ between groups.

### Conclusion

This study provides the first morphometric data on fabella in Turkish cadavers and suggests no significant relationship between fabella presence and knee osteoarthritis.

**Data availability statement:** All relevant anonymized data (available in the Supporting Information files: Cadaver.xlsx, 2020-2021 prevalence control.xlsx, and 2020-2021 prevalence patient.xlsx) are provided.

**Funding:** The budget of this research was covered by the Scientific Research Projects Coordination Unit of the University of Health Sciences within the scope of the project numbered 2023/022. The funders had no role in study design, data collection and analysis, decision to publish, or preparation of the manuscript.

**Competing interests:** The authors have declared that no competing interests exist.

While fabella is a common anatomical variant, its clinical relevance in osteoarthritis remains inconclusive, warranting further investigation.

## Introduction

The fabella is a sesamoid ossicle typically located within the tendon of the lateral head of the gastrocnemius muscle. However, it has been demonstrated that it can also be found in the tendon of the medial head, although rarely. The fabella contributes to minimizing frictional injury to the tendon, enhances the mechanical efficiency of the gastrocnemius muscle, and plays a role in stabilizing the posterolateral aspect of the knee joint through its interaction with the fabellofibular ligament [1–4].

In the literature, fabella prevalence is generally reported to be higher in Asian populations, and recent studies further support that it is particularly common in Asian countries [3,5–7]. This is thought to be due to cultural differences. Kneeling and squatting are common in Asian societies. These postures subject the fabellar region to sustained mechanical compression against the posterior aspect of the lateral femoral condyle [8–10]. According to Wolff's law of bone adaptation, this repetitive stress initiates and accelerates the process of endochondral ossification at this site, effectively reinforcing the tendon as a sesamoid bone to improve its mechanical function under load [11]. Therefore, the sustained mechanical stress generated by these cultural postures provides a biological basis for the more frequent and more prominently ossified appearance of the fabella observed in Asian populations.

Fabella is usually detected incidentally on radiographs. Yet, depending on its location or osteophytization, fabella may cause atypical pain in the posterolateral region of the knee. Clinically, it leads to pathologies such as fracture, fabella syndrome, nervus fibularis communis paralysis, chondromalacia, and osteoarthritis [7,10,12].

In recent years, it has been revealed that the relationship between fabella and clinical conditions is significant [6,13]. For this reason, studies on fabella and clinical conditions have started to increase. Undoubtedly, the most striking of these clinical conditions is the relationship between knee osteoarthritis and the presence of fabella, which is still unclear. In Prittchett's study, when he compared the radiographs of individuals with osteoarthritis in the knee and normal individuals, he found that fabella was more likely in the osteoarthritic knee [14].

The purpose of this study was to (1) define the morphometric properties of the fabella via cadaveric dissection and (2) investigate its prevalence, lateralization pattern, gender distribution, and association with knee osteoarthritis using radiological examination in the Turkish population. This study aimed to address the lack of literature data on this subject specific to the Turkish population.

## Materials and methods

This study was found ethically appropriate according to the Decision number 2023/83 dated 14.03.2023 of the Gülhane Scientific Research Ethics Committee of the University of Health Sciences. The need for informed consent was waived by the

Gülhane Scientific Research Ethics Committee because the radiological data were analyzed retrospectively, and no personally identifiable information was accessed by the authors.

Cadaveric dissections were carried out in the Anatomy Laboratory of Gülhane Faculty of Medicine between 20/03/2023 and 27/03/2023. The specimens were delivered to the Department of Histology and Embryology on 29/03/2023, and the histological analysis results were reported to the researchers on 29/05/2023.

For the radiological study, retrospective knee radiographs covering the period between 01/01/2020 and 31/12/2021 were included. These data were accessed and analyzed between 20/03/2023 and 20/05/2023.

## Cadaveric study

In our study, 32 formaldehyde-fixed cadavers (six females, 26 males) and three male leg specimens (two left legs, one right leg) aged 60 to –85 years were dissected in the anatomy dissection laboratory of Gülhane Medical Faculty, University of Health Sciences. The right legs of two female cadavers were amputated. The sex of each cadaver and isolated limb was obtained from institutional anatomical inventory records linked to identification numbers attached to the specimens. No cadaver with unknown sex was included in the study. The lateral and medial heads of the gastrocnemius muscle were identified by dissection in the posterior region of the knee and fibrocartilaginous structures were investigated. In suspicious cases, the capsule was incised and condyles of the femur were exposed. The excised fabellae were stored in 10% formaldehyde solution and separated from the surrounding tissues under a dissecting microscope (Carl ZEISS, Germany; 4.0–6.0×magnification). Superior Inferior (SI)-Anterior Posterior (AP) dimensions were measured with digital calipers by

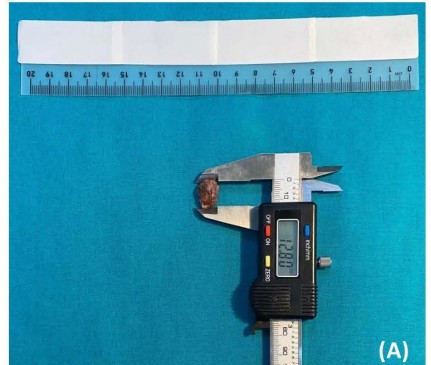
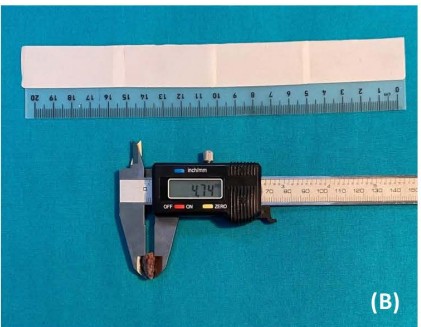
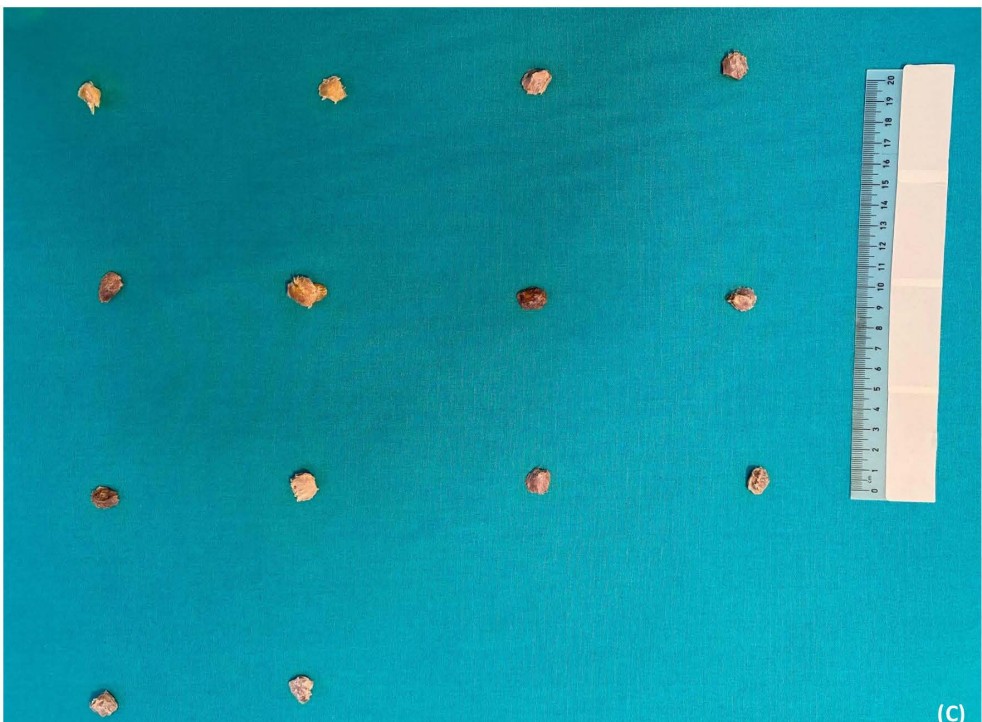

**Fig 1. Fabella measurements were performed by two-dimensionally.** (A) SI measurement with caliper. (B) AP measurement with caliper. (C) Collective view of the removed Fabella.

two independent investigators (Fig 1). The specimens were then transferred to the Gülhane Medical Faculty Department of Histology and Embryology for histopathologic analysis.

The fabellae were decalcified in 20% EDTA solution for eight days. After decalcification, 7-µm-thick sections were prepared using a cryotome and dried on slides overnight. Histological evaluation was performed using Hematoxylin-Eosin and Masson's Trichrome staining protocols.

### Radiologic study

The radiologic part of the study was conducted using the image database of the Department of Radiology of Gülhane Training and Research Hospital, Ankara, Türkiye, between the dates of January 2020 and December 2021. Radiological data from this period were retrospectively analyzed, and no personally identifiable information was accessed by the authors. Patients group composed of patients over 40 years, had bilateral (anterior-posterior/lateral) knee roentgenograms, and were diagnosed with knee osteoarthritis. The control group composed of radiology department admissions with the same age group and had bilateral knee roentgenograms throughout the same time period. Patients with suboptimal image quality, non-standard posture, unilateral images, and the presence of duplicate records were excluded. During the initial assessment, 2246 patients and 3136 healthy individuals were identified, Consequently, 1712 patients and 2304 controls were enrolled in the study. In all cases, the presence of the fabella was independently confirmed by two radiologists and two anatomists (Fig 2).

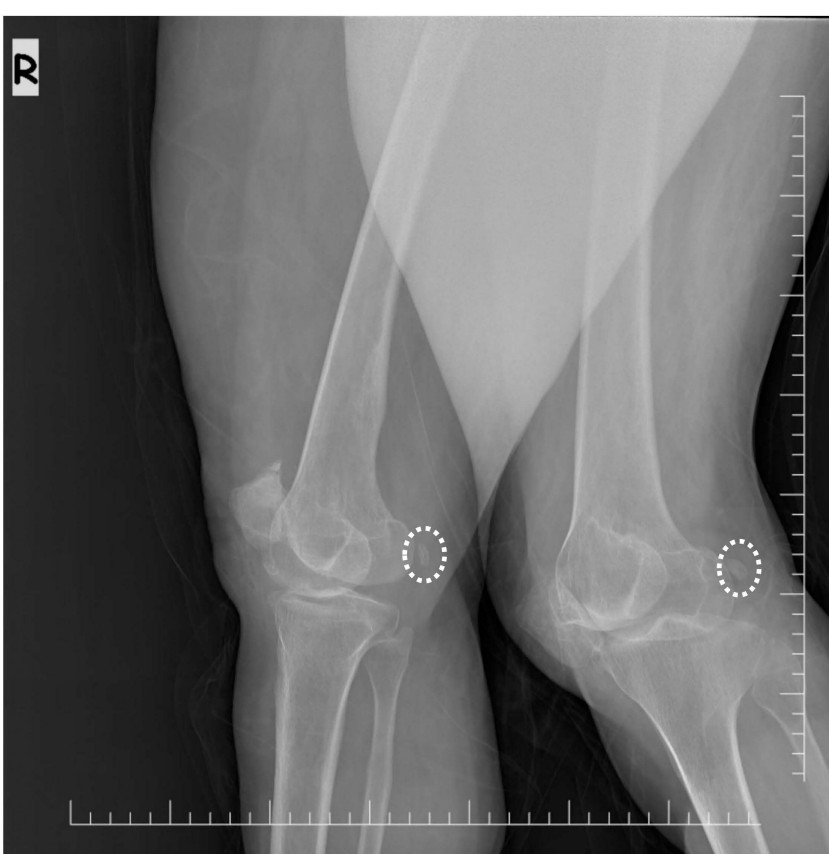

**Fig 2. A bilateral lateral knee roentgenogram showing the fabellae, each surrounded by an elliptical circle, in both the right and left legs.**

## Statistical analysis

In the cadaver study, descriptive statistics were evaluated concerning gender, the number of fabellae, measurements, and categorical data presented as frequency (n) and percentage (%). In the radiology study, calculated the sample size based on a statistical power of 80% and a Type 1 error of 5%. The normal distribution of continuous variables was analyzed by Kolmogorov-Smirnov (n > 50) and Skewness-Kurtosis tests. Independent Samples T-test was applied for continuous variables in comparisons between groups, and Chi-square ($\chi^2$) test was used to evaluate the relationships between categorical variables. The statistical significance level was taken as $p < 0.05$ in the calculations and SPSS (IBM SPSS for Windows, version 26.0 software) statistical package program was used for the analyses.

## Results

### Cadaveric study

Since amputee legs and specimen tissues were in our sample, the prevalence calculation was based on the total number of knees. Fourteen fabella, all of which were excised from the tendon of the lateral head of the gastrocnemius muscle were sent for histopathologic examination. It was determined that two of these structures were muscular tissue and 12 were bony fabellae (Fig 3). Therefore, the measurement results of 12 fabella were included to statistical analysis. During dissections, traces of three fabellae were found on the lateral condyle of the femur. These traces were named sesamoid traces made by the fabella (Fig 4).

In 65 dissected knees, 12 fabella were found. The total prevalence of fabella was 18.4% (95% CI: 9.8% to 30.0%). Three fabella were found in 27 male right legs and one fabella was found in 4 female right legs. Six fabella were found in 28 male left legs and two fabella were found in 6 female left legs. The prevalence of fabella between males and females was not statistically significant (p > 0.05). When the SI and AP measurements of the right and left fabella were analyzed, although the fabella measurements of female were slightly higher than those of men, there was no significant statistical difference between the SI and AP measurements of male and female (p > 0.05).

### Radiologic study

The study was conducted with data collected from 1712 patients (42.6%) and 2304 healthy individuals (57.4%) who were admitted to the Radiology Clinic of Gülhane Training and Research Hospital, University of Health Sciences. In the evaluation of the participants, the gender distribution of the healthy and patients; for the patient group; 79.2% (n: 1356) of the

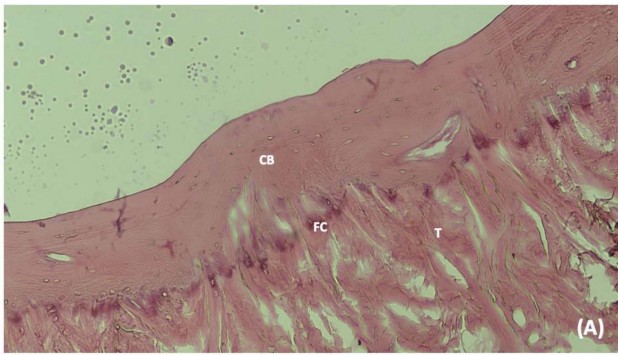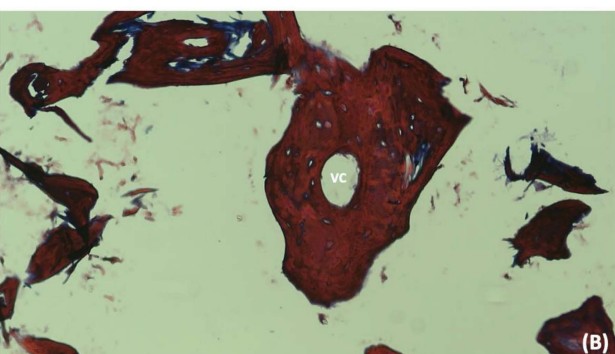

**Fig 3. Representative histological sections of the fabella.** (A) H&E-stained fabella section showing the tendon (T) attachment to compact bone (CB) with a fibrocartilage (FC) transition zone. (B) Trabecular bone within the fabella, stained with Masson's trichrome, revealing a central vascular canal (VC) that facilitates nutrient supply to bone tissues.

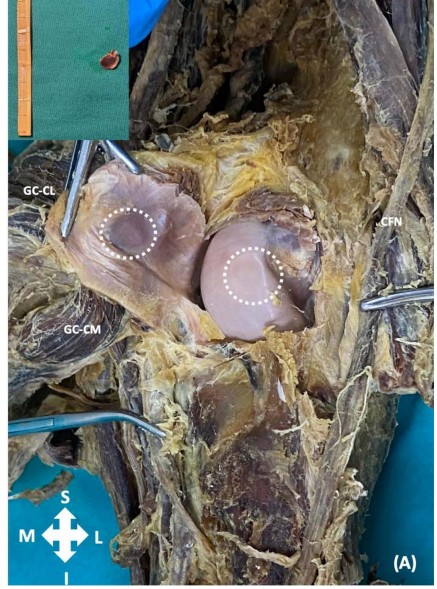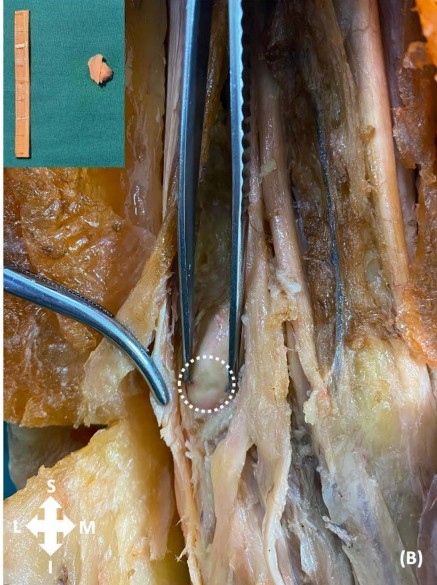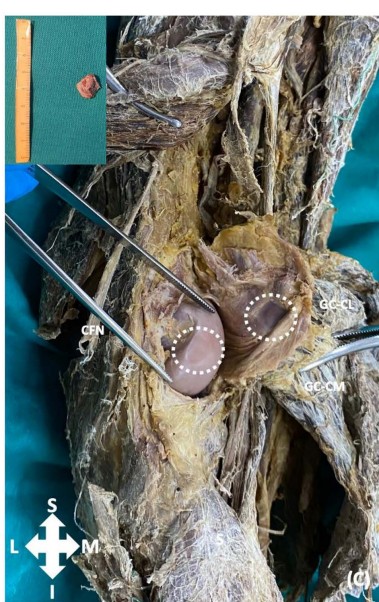

**Fig 4. Fabella sesamoid traces with fabellas.** (A) Medial head of the Gastrocnemius muscle (GC-CM) and lateral head of the Gastrocnemius muscle (GC-CL) in the right leg of a mature male cadaver, with the common fibular nerve (CFN) visible. (B) The left leg of a mature female cadaver. (C) GC-CM and GC-CL in the left leg of a mature female cadaver, with the CFN and Soleus muscle (S) visible.

study were female and 20.8% (n: 356) were male, while in the control group; 75.2% (n: 1733) of the study were female and 24.8% (n: 571) were male. The mean±standard deviation of the age of the participants was for the patient group; 63.48±9.95 (minimum age: 40 years, maximum: 96 years, Median: 63), for the control group; 58.84±11.06 (minimum age: 40 years, oldest age: 98, Median value: 58). The total prevalence of fabella was 36.5% (95% CI: 35.0% to 38.0%), the prevalence of female fabella was 35.6% (95% CI: 33.9%–37.3%), and the prevalence of male fabella was 39.5% (95% CI: 36.4%–42.6%). No statistically significant sex-based difference was observed in the prevalence of the fabella (p>0.05). Table 1 summarizes the distribution of fabella by sex.

The presence of fabella on the right and left legs of the patients and healthy individuals is given in Table 2 and the rate of presence of fabella is 37.6% (95% CI: 35.3% to 39.9%) in patients and 35.7% (95% CI: 33.8% to 37.6%) in healthy individuals. The prevalence of fabella in patient individuals did not differ significantly from those who were healthy (p>0.05). Statistical analysis revealed no significant difference between the groups in terms of laterality (side) (p>0.05). In other words, we can say that the one-sided or double-sided fabella variable does not change according to patients or healthy people. Table 2 presents the distribution of unilateral and bilateral fabellae, as well as right- and left-sided occurrence, in patient and healthy individuals.

**Table 1. Prevalence of Fabella for Female and Male.**

|  |  | Gender |  |  |  |  |  | Chi-square |
|---|---|---|---|---|---|---|---|---|
|  |  | Female |  | Male |  | Total |  |  |
|  |  | n | % | n | % | n | % | p |
| Prevalence of Fabella | The presence of a fabella in any leg | 1099 | 35,6 | 366 | 39,5 | 1465 | 36,5 | 0.463 |
|  | Absent | 1990 | 64,4 | 561 | 60,5 | 2551 | 63,5 |  |
|  | Total | 3089 | 100,0 | 927 | 100,0 | 4016 | 100,0 |  |

**Table 2. Presence of fabella on the right and/or left legs of patient and healthy individuals.**

| | | Patient | | Healthy | | Total | | Chi-squared test | |
|---|---|---|---|---|---|---|---|---|---|
| | | n | % | n | % | n | % | χ² | p |
| Fabella in the Right Leg | Absent | 1290 | 75,4 | 1723 | 74,8 | 3013 | 75,0 | 0,169 | 0,681 |
| | Present | 422 | 24,6 | 581 | 25,2 | 1003 | 25,0 | | |
| Fabella in the Left Leg | Absent | 1181 | 69,0 | 1630 | 70,7 | 2811 | 70,0 | 1,453 | 0,228 |
| | Present | 531 | 31,0 | 674 | 29,3 | 1205 | 30,0 | | |
| Fabella | Absent | 1069 | 62.4 | 1482 | 64.3 | 2551 | 63.5 | 1.50 | 0.221 |
| | Present | 643 | 37.6 | 822 | 35.7 | 1465 | 36.4 | | |
| Side | Unilateral | 333 | 19.4 | 389 | 16.9 | 722 | 18 | 2.88 | 0.090 |
| | Bilateral | 310 | 18.1 | 433 | 18.8 | 743 | 18.5 | | |

## Discussion

### Cadaveric study

Located within the tendon of the lateral head of the gastrocnemius muscle, the fabella is a sesamoid bone that generally remains asymptomatic and is often discovered incidentally during imaging. The methods used in prevalence studies significantly affect the results, as X-ray–based techniques may overlook cartilaginous fabellae, leading to differences in reported rates [15]. In the dissection study conducted by Zeng et al., 11 of 19 the cartilaginous fabella identified by palpation could not be visualized on radiographs [2]. Chew et al. likewise showed that cartilage fabellae visible on MRI were not detected radiographically [8].

In cadaveric studies, the prevalence of fabella was generally found to be higher, with the highest rates reported especially in Asian populations. Table 3 provides cadaver-based fabella prevalence values previously reported in the literature.

Our results regarding fabella prevalence (18.4%), indicating no gender association, are consistent with the rate reported by Phukubye et al. (11.7%) [20]. However, they differ markedly from the much higher rates reported in East Asian populations, such as the 68.6% reported by Tabira et al. [3]. This highlights how ethnic and geographic factors can lead to substantial variation in fabella prevalence. To our knowledge, this is the first study to present cadaveric fabella prevalence statistics exclusively from a Turkish community.

All fabellae identified in our study were located exclusively in the tendon of the lateral head of the gastrocnemius, contrary to reports of medial location in the literature. This result is consistent with Phukubye et al. [20] but contrasts with the reports of medial location by Zeng et al. [2] and Kawashima et al. [1].

**Table 3. Cadaver study-based prevalence values reported in the literature.**

| Researchers | Year | Population | Method | Age | Fabella/Knee | Prevalence |
|---|---|---|---|---|---|---|
| Chihlas et al. [16] | 1993 | United States | Cadaver | – | 18/78 | 27% |
| Minowa et al. [17] | 2004 | Japan | Cadaver | 54-98 | 182/212 | 85.8% |
| Kawashima et al. [1] | 2007 | Japan | Cadaver | 66-100 | 99/150 | 66.0% |
| Raheem et al. [18] | 2007 | Ireland | Cadaver | 84±8.1 | 2/22 | 9.1% |
| Silva et al. [19] | 2010 | Brazil | Cadaver | 38-78 | 2/62 | 3.1% |
| Phukubye and Oyedele [20] | 2011 | Africa | Cadaver | 40-95 | 12/102 | 11.7% |
| Tabira et al. [3] | 2012 | Japan | Cadaver | 74.5±12.3 | 70/102 | 68.6% |
| Piyawinijwong et al. [21] | 2012 | Thailand | Cadaver | 30-97 | 144/372 | 38.7% |
| Mohite et al. [22] | 2016 | India | Cadaver | – | 5/40 | 12.5% |
| Present Study | 2023 | Türkiye | Cadaver | 60-85 | 12/65 | 18.4% |

In the lateral condyle region of the femur, traces of sesamoid bones were detected. Kawashima et al. reported that the bony fabella left trace on the femoral condyle [1]. Tabira et al. found a round depression in this region [3]. Zeng et al. reported that bony and cartilaginous fabellae formed traces on the femoral condyle [2].

Although the SI and AP measurements of the fabella were numerically higher in females, this difference was not statistically significant (p > 0.05). The most likely reason for this finding is that there were only six female cadavers in our collection. When there are few samples, individual differences, such as one or two people naturally having larger fabellae, can alter the average and create the impression that the population is different to what it really is. Accordingly, this apparent increase cannot be considered evidence of genuine sexual dimorphism. This conclusion is supported by the work of Chew et al., who also found no significant correlation between gender and fabella dimensions [8]. In summary, our data suggest that fabella size is primarily influenced by factors independent of sex, such as genetic predisposition and localised biomechanical stress [15].

## Radiology study

The prevalence of fabella exhibits variation across populations, influenced by factors such as ethnicity and geographical distribution. Moreover, this prevalence may also differ within distinct regions of the same country. Table 4 summarizes radiology-based prevalence values from different populations worldwide.

**Table 4. Radiology study-based prevalence values reported in the literature.**

| Researchers | Year | Population | Method | Age | Fabella/Knee | Prevalence |
|---|---|---|---|---|---|---|
| Yu et al. [23] | 1996 | United States | Radiography (MRI) | 12-72 | 19/100 | 19.0% |
| Sarin et al. [24] | 1999 | United States | Radiography (Direct Radiography) | 19-84 | 55/224 | 31.3% |
| Zeng et al. [2] | 2012 | China | Cadaver+ Radiography | 64-83 | 53/61 | 86.9% |
| Chew et al. [8] | 2014 | Singapore | Radiology (Knee arthroscopy, MRI) | 14-55 | 25/80 | 31.3% |
| Ehara [25] | 2014 | Japan | Radiography (MRI) | 4-89 | 200/653 | 30.6% |
| Hauser et al. [26] | 2015 | Switzerland | Radiology (CT) | 20-104 | 105/400 | 26.3% |
| Egerci et al. [4] | 2016 | Türkiye | Radiography (Direct Radiography) | 18-90 | 228/1000 | 22.8% |
| Ghimire et al. [27] | 2017 | Nepal | Radiography (Direct Radiography) | – | 19/155 | 12.3% |
| Hedderwick et al. [28] | 2017 | New Zealand | Radiology (MRI) | Mean age 23 ± 3.2 years | 8/28 | 28.5% |
| Pop et al. [29] | 2018 | Romania | Radiology (MRI) | 21-43 | 73/862 | 8.4% |
| Berthaume et al. [30] | 2019 | Korea | Radiology (CT) | 21-60 | 94/212 | 44.34% |
| Hou et al. [10] | 2019 | China | Radiography (Direct Radiography) | 21months-86 years | 660/1359 | 48.6% |
| Hur et al. [13] | 2020 | Korea | Radiography (Direct Radiography) | 18+ | 2172/4252 | 51.1% |
| Adedigba et al. [9] | 2021 | Africa | Radiography (Direct Radiography) | 3-100 | 77/754 | 10.2% |
| Sari et al. [6] | 2021 | Türkiye | Radiography (Direct Radiography) | Female Mean±SD 61.33 ± 12.38 Male Mean±SD 59.28 ± 15.25 | 380/2000 | 19% |
| Matroushi et al. [31] | 2021 | Omani | Radiography+MRI | Radiography: 20–100 MRI: ≥ 20, < 45 | 196/813 | 24.1% |
| Akkoç et al. [32] | 2022 | Türkiye | Radiology (MRI) | 18+ | 605/2035 | 29.7% |
| Zhong et al. [33] | 2022 | Korea | Radiology (MRI) | 8-91 | 402/1011 | 39.8% |
| Present Study | 2023 | Türkiye | Radiography (Direct Radiography) | 40-98 | 1465/8032 | 18.2% |
| Packirisamy et al. [34] | 2024 | Saudi Arabia | Radiology (MRI) | 20-90 | 170/820 | 20.73% |
| Shete et al. [35] | 2024 | India | Radiography (Direct Radiography) | 18+ | 87/500 | 17.40% |

Radiological studies conducted on the Turkish population reveal notable variations in fabella prevalence. The existing literature reports a general prevalence of approximately 20–39% [4,6,32,36], which may indicate regional differences within the country. In particular, Akkoç et al. found relatively high and similar prevalence values [32], and our study diverge from the results of other studies. The consistent lack of a significant difference between sexes observed across all these studies further strengthens our finding that fabella presence is independent of gender in the Turkish population.

In line with previous studies conducted in Turkey, we did not observe a statistically significant association between sex and fabella prevalence [4,6,13,36,37]. Nevertheless, a recent meta-analysis reported a slightly higher fabella prevalence in males globally [15]. Therefore, our findings suggest that any sex-related differences, if they exist, may be population-specific rather than universal.

Our study found no statistically significant association between the presence of a fabella and knee osteoarthritis (prevalence difference 1.9%; 95% CI −1.1% to 4.9%), consistent with a confidence interval that includes zero. Our findings contradict the positive associations reported in the meta-analysis by Asghar et al. [38] and in the study by Pritchett [14]. Biomechanical explanations may account for the lack of a significant relationship in our study. Although the literature suggests that the relationship between the fabella and osteoarthritis may develop through changes in knee mechanics, such as valgus alignment, shifting the load to the lateral compartment and potentially increasing stress on the fabella [6], this relationship may be dependent on the severity of osteoarthritis. The pathological consequences of such mechanical interactions (marked narrowing of the joint space, multiple osteophyte formation, subchondral sclerosis, etc.) are typically evident in advanced osteoarthritis [39]. These advanced changes provide the mechanical stimuli that trigger ossification or enlargement of the fabella [10]. If the prevalence of advanced osteoarthritis is low in our study population, this could explain why no association was found between the fabella and osteoarthritis. This suggests that the relationship between the fabella and osteoarthritis may depend on the disease's stage.

In our study, the frequencies of unilateral and bilateral fabella did not show significant differences between patients with osteoarthritis and healthy individuals. Laterality analysis indicated that right-sided fabellae were more frequently bilateral, whereas left-sided fabellae tended to appear more often as unilateral. Sari et al. reported that unilateral or bilateral fabella on the left side was more common in patients with osteoarthritis [6]. Asghar et al.'s meta-analysis showed that, overall, bilateral fabella is more prevalent than unilateral, and that fabella tends to occur more often on the right side; however, these laterality patterns were not analyzed separately for osteoarthritic knees [38]. Disparities among studies are probably attributable to variances in population characteristics, predominant limb utilization, and asymmetric mechanical loads.

The potential functional roles of the fabella should be considered, in addition to its prevalence and morphology. As noted in the introduction, it may reduce tendon friction, improve the mechanical advantage of the gastrocnemius, and contribute to posterolateral knee stability through the fabellofibular ligament [1–4]. Theoretically, the absence of the fabella or the fabellofibular complex could diminish these effects, particularly posterolateral stability [1,17]. Similarly, the loss of its pulley-like action may reduce the gastrocnemius muscle's mechanical efficiency and increase the tendon's susceptibility to friction-induced stress [40]. Whereas the absence of a fabella is considered a normal anatomical variation, compensatory mechanisms within the musculoskeletal system likely prevent functional deficits in most individuals. Nevertheless, while the fabella can serve as a biomechanical aid, its absence does not necessarily indicate dysfunction. Rather, its absence reflects variability in knee biomechanics and susceptibility to posterolateral pathology.

This research is limited at both the cadaveric and radiological stages. Particularly the fabellae identified through manual palpation could not be confirmed through radiology at the cadaveric stage, which would have strengthened the evidence for their presence. Unfortunately, it was not feasible to perform radiographic imaging on the cadavers. In the radiological part of the study, the use of radiography alone meant that non-ossified cartilaginous fabellae could not be detected. Thus, our findings reflect only the prevalence of ossified fabellae. Although magnetic resonance imaging (MRI) would allow for a more detailed evaluation, the cost-effectiveness of using it for this study is debatable.

 

## Conclusion

The fabella, a sesamoid bone of the knee joint, continues to attract anatomical interest due to its variable presence, morphology, and clinical implications. To our knowledge, this is the first study in Türkiye to document the prevalence and morphological characteristics of cadaveric fabellae using histological staining. Notably, our radiological findings revealed no significant association between fabella presence and knee osteoarthritis—a result that contrasts with several previous reports. This divergence highlights the complexity of the fabella's role in joint pathology and underscores the necessity of broader, population-specific investigations to elucidate its clinical relevance more accurately.

## Supporting information

**S1 File. 2020–2021 prevalence control.**
(XLSX)

**S2 File. 2020–2021 prevalence patient.**
(XLSX)

**S3 File. Cadaver.**
(XLSX)

## Acknowledgments

This study was produced from the medical specialization thesis titled 'Morphometry of Fabella in Human Cadaver and Radiological Evaluation of Fabella in Osteoarthritis Cases' which was conducted under the supervision of Assoc. Prof. Nurcan Ercıktı, with co-researchers from different medical disciplines who are experienced in their fields, and defended by Dr. Ebru Yolaçan on 15.06.2023.

## Author contributions

**Conceptualization:** Ebru Yolaçan, Necati Salman.

**Data curation:** Mehmet Ali Güner, Mislina Utlu, Laçin Ramazanlı.

**Formal analysis:** Barış Baykal, Ayşe Özdemir.

**Investigation:** Ebru Yolaçan, Barış Baykal, Mislina Utlu.

**Methodology:** Nurcan Ercıktı, Barış Baykal.

**Project administration:** Barış Baykal, Uğur Bozlar, Necdet Kocabıyık.

**Software:** Mehmet Ali Güner.

**Supervision:** Nurcan Ercıktı.

**Validation:** Ebru Yolaçan, Nurcan Ercıktı.

**Visualization:** Ebru Yolaçan, Barış Baykal, Uğur Bozlar, Ayşe Özdemir.

**Writing – original draft:** Ebru Yolaçan.

**Writing – review & editing:** Ebru Yolaçan, Barış Baykal, Uğur Bozlar, Necdet Kocabıyık, Mehmet Ali Güner, Necati Salman, Ayşe Özdemir, Mislina Utlu, Laçin Ramazanlı.

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
