## [Decision Letter · Decision Letter 0]

4 Nov 2025

PONE-D-25-50561PREVALENCE AND MORPHOMETRY OF THE FABELLA IN CADAVERIC AND RADIOLOGICAL STUDIES: IS THERE A LINK TO KNEE OSTEOARTHRITIS?PLOS ONE?

Dear Dr. YOLACAN,

Thank you for submitting your manuscript to PLOS ONE. After careful consideration, we feel that it has merit but does not fully meet PLOS ONE’s publication criteria as it currently stands. Therefore, we invite you to submit a revised version of the manuscript that addresses the points raised during the review process.

We look forward to receiving your revised manuscript.

Kind regards,

Srinivasa Rao Sirasanagandla

Academic Editor

PLOS ONE

Journal Requirements:

“The budget of this research was covered by the Scientific Research Projects Coordination Unit of the University of Health Sciences within the scope of the project numbered 2023/022.”

“NO authors have competing interests”

5. Please include captions for your Supporting Information files at the end of your manuscript, and update any in-text citations to match accordingly. Please see our Supporting Information guidelines for more information: http://journals.plos.org/plosone/s/supporting-information .

Additional Editor Comments:

The manuscript is well written and clinically relevant.

Consider adding the following published article on the Omani population in the table and discussion section.

Al Matroushi OD, Sirasanagandla SR, Al Shabibi A, Al Obaidani A, Al Dhuhli H, Jaju S, Al Mushaiqri M. Radiological study of fabella in Omani subjects at a tertiary care center. Anatomy & Cell Biology. 2021 Sep 30;54(3):315-20.

Correct the grammar and language throughout the manuscript.

Reviewers' comments:

Reviewer's Responses to Questions

**Comments to the Author**

1. Is the manuscript technically sound, and do the data support the conclusions?

Reviewer #1: Yes

Reviewer #2: Yes

2. Has the statistical analysis been performed appropriately and rigorously?

Reviewer #1: Yes

Reviewer #2: Yes

3. Have the authors made all data underlying the findings in their manuscript fully available?

Reviewer #1: Yes

Reviewer #2: Yes

4. Is the manuscript presented in an intelligible fashion and written in standard English?

Reviewer #1: Yes

Reviewer #2: Yes

Reviewer #1: 1.Reference introduction first paragraph, If fabella is absent do you think the above functions are compromised?

2.In the literature, it is generally reported to be more common in Asia [5-7]. Current studies in

the literature indicate that fabella is common in Asian countries. Merge these sentences

3.while the radiological part aimed to examine the prevalence of fabellae, lateralization pattern, genders distribution replace genders with gender

4. In the 2.1 section, In our study, 32 formaldehyde-fixed cadavers (6 six females, 26 males) and 3 three male

specimens (How do you ascertain the sex by leg ) (2 two left legs, 1 one right leg) aged between 60 to 85 years were dissected in the anatomy

5. The right legs of both (two) female cadavers were amputated.

6. gastrocnemius muscle were palpated (identified) by dissection in the posterior region of the knee

7. The fabellae were decalcified in 20% EDTA solution for 8 (eight) days

8. It was determined that 2 (two) of these structures were muscular tissue

9. The prevalence of fabella in patients (patient) individuals did not differ significantly

10. Table 3 last column should be prevalence

11. In our work, due to comparing fabella measurements according to gender, the right

fabella length (SI) was found to be 10.46±1.19 mm in males and 10.90 mm in females. Right

fabella thickness (AP) was 4.23±1.42 mm in males and 3.95 mm in females. Left fabella SI was

10.83±0.68 mm in males and 12.45±1.39 mm in females; left fabella AP was 4.02±0.69 mm in

males and 4.49±0.77 mm in females (n general females bone are said to be smaller than males. How would you justify the reason for high values in female

11. Table 4 last column should be prevalence

Reviewer #2: Kindly got through the detailed review remarks and make necessary changes and resubmit

Review report:

PREVALENCE AND MORPHOMETRY OF THE FABELLA IN CADAVERIC AND

RADIOLOGICAL STUDIES: IS THERE A LINK TO KNEE OSTEOARTHRITIS?

The prevalence and morphometry of the fabella in a Turkish population, as well as any possible connections to osteoarthritis (OA), are discussed in this study that combines cadaveric, histological, and radiographic investigations.

To improve readability, flow, and clarity, however, some small changes are required, particularly in the Introduction, Results, and Discussion sections. Minor terminology and statistical expression errors should be fixed, and certain phrases can be made shorter.

Introduction:

• Condense the repeated statements about fabella prevalence and Asian populations.

• Clarify cultural reasoning: why the higher prevalence in Asian populations is thought to be associated with habitual kneeling and squatting postures and which may increase mechanical stimulation and ossification of the fabella.”

• Last paragraph can be ended with more clearer research aim /objectives

Results

• Provide confidence intervals (if available) for key prevalence figures.

• Add a brief linking statement before each table to improve flow, For example-Table 1 summarizes the distribution of fabella by sex.

• Use table only legends when it is shown in the table (such as asterisks in table-2 used in the legend, but it is not shown in the content values)

• In table -3, use of symbol % - should be in consistent with that of text. Generally it is used after number (eg. 14%.... ) Kindly check this

Discussion

Some paragraphs repeat details from the Results. Avoid repetition to make it concise and consider to condense to focus on interpretation.

You may add a paragraph explicitly discussing possible biomechanical explanations for the lack of association between fabella and OA.

**Do you want your identity to be public for this peer review?** For information about this choice, including consent withdrawal, please see our Privacy Policy

Reviewer #1: No

Reviewer #2: No

---

## [Author Response · Author response to Decision Letter 1]

11 Dec 2025

Dear Editor,

Manuscript Title: Prevalence and Morphometry of the Fabella in Cadaveric and Radiological Studies: Is There a Link to Knee Osteoarthritis?

Corresponding Author: Ebru YOLAÇAN, MD

Thank you for handling our manuscript and for forwarding it to the reviewers. Our point-by-point responses to the reviewers' comments are provided below. All changes and additions in the revised manuscript have been highlighted. In line with the editor’s recommendation, the study by Al Matroushi et al. (2021) (Ref.31) has also been incorporated into the relevant comparative table.

Dear Reviewer #1 and Reviewer #2,

Thank you for your valuable time and for your insightful and constructive critiques of our manuscript. In our commitment to improving the methodological rigor and clinical relevance of our study, we have carefully considered all of your suggestions and have implemented comprehensive revisions accordingly.

Our point-by-point responses to your comments are presented below.

Revisions made in response to Reviewer #1 are highlighted in yellow, and revisions made in response to Reviewer #2 are highlighted in light green. As we did not use line numbers in the revised manuscript, the changes have been indicated directly within the relevant sections using the specified highlight colors.

CORRECTIONS:

Authors’ response to Reviewer: 1

1. Reference introduction first paragraph, If fabella is absent do you think the above functions are compromised?

Author’s Response:

Dear Reviewer, thank you so much for your helpful question and feedback about what it means when the fabella is missing. We agree that this is an essential point that gives our results a lot of clinical context. We have completely rewritten our manuscript to address this point in response to your question. We didn't directly measure the fabella's function because our study was only on its prevalence and shape. We have now improved the Discussion section by adding a new paragraph that looks at what would happen if it is not present, based on a synthesis of the current literature. The new paragraph suggests that not having the fabellofibular complex could make the posterolateral stability less stable. It also says that losing its pulley-like action could make the gastrocnemius less mechanically efficient and more likely to get stressed by friction. We have backed up these theoretical statements by the right citations from the literature [1, 17, 40], which is very important. We also want to stress that the fabella is not always present in people, and that the body's compensatory mechanisms imply that this does not indicate that most people have functional failure. We think that adding this immediately addresses the interesting topic you mentioned and makes our article much more relevant to the conversation and the clinic.

2. In the literature, it is generally reported to be more common in Asia [5-7]. Current studies in the literature indicate that fabella is common in Asian countries. Merge these sentences

Author’s Response:

Thank you for this helpful suggestion. The two sentences have been merged to improve clarity and avoid repetition. The revised sentence now reads as follows:

“In the literature, fabella prevalence is generally reported to be higher in Asian populations, and recent studies further support that it is particularly common in Asian countries [3, 5–7].”

The change has been incorporated into the revised manuscript.

3. While the radiological part aimed to examine the prevalence of fabellae, lateralization pattern, genders distribution replace genders with gender

Author’s Response:

Thank you for pointing this out. The term “genders distribution” has been corrected to “gender distribution” in the revised manuscript.

4. In the 2.1 section, In our study, 32 formaldehyde-fixed cadavers (6 six females, 26 males) and 3 three male specimens (How do you ascertain the sex by leg ) (2 two left legs, 1 one right leg) aged between 60 to 85 years were dissected in the anatomy.

Author’s Response:

Thanks for these helpful suggestions. The suggested changes regarding the phrasing, numerical style, and terminology have been provided in the revised paper. As suggested, the words "six," "three," "two," and "one" have been changed, and the phrase "leg specimens" has been added to make things clearer. Your note also says that the age range has been changed to "aged 60 to 85 years." We would like to go into more information about how sex was determined for the isolated limb specimens in response to your question. The isolated lower-limb specimens individually did not determine sex. All of the bodies and limbs we have in our lab have identifying numbers attached to the auricle or great toe. These numbers match the official anatomical inventory records of the institution. These documents show the person's sex, age, year of body donation, and cause of death. Consequently, the sex of the three isolated lower-limb specimens was directly derived from these institutional data. Moreover, despite being classified as “leg samples,” the proximal soft tissues were well preserved to permit visibility of the genital region, offering a secondary anatomical confirmation that was entirely congruent with the inventory data. The study did not include any specimens whose sex was unknown or could not be determined. This explanation has now been added to the Materials and Methods section to make sure there is no confusion. All of the suggested changes have been made, and the new version shows these changes. Thanks again for your helpful feedback and thorough assessment.

5. The right legs of both (two) female cadavers were amputated.

Author’s Response:

We thank the reviewer for pointing this out. The sentence has been revised for clarity as suggested and now reads:

"The right legs of two female cadavers were amputated."

6. gastrocnemius muscle were palpated (identified) by dissection in the posterior region of the knee

Author’s Response:

We thank the reviewer for this precise and accurate correction. The sentence has been rephrased as recommended and now reads:

"The lateral and medial heads of the gastrocnemius muscle were identified by dissection in the posterior region of the knee and fibrocartilaginous structures were investigated."

7. The fabellae were decalcified in 20% EDTA solution for 8 (eight) days

Author’s Response:

The phrase has been corrected to ‘for eight days’ in accordance with the recommended numerical style, and consistency has been ensured throughout the manuscript.

8. It was determined that 2 (two) of these structures were muscular tissues

Author’s Response:

The expression has been revised as ‘two of these structures’ to maintain consistency in numerical style across the manuscript.

9. The prevalence of fabella in patients (patient) individuals did not differ significantly

Author’s Response:

The wording has been corrected to ‘in patient individuals’ in line with the reviewer’s recommendation.

10. Table 3 last column should be prevalence

Author’s Response:

Thank you for bringing this to our attention. The heading of the last column in Table 3 has been revised to ‘Prevalence’ as suggested.

11. In our work, due to comparing fabella measurements according to gender, the right fabella length (SI) was found to be 10.46±1.19 mm in males and 10.90 mm in females. Right fabella thickness (AP) was 4.23±1.42 mm in males and 3.95 mm in females. Left fabella SI was 10.83±0.68 mm in males and 12.45±1.39 mm in females; left fabella AP was 4.02±0.69 mm in males and 4.49±0.77 mm in females (n general females bone are said to be smaller than males. How would you justify the reason for high values in female

Author’s Response:

Thank you for this thoughtful observation. We carefully reviewed the sex-based morphometric data and provide the following clarification in the revised manuscript.

Although some fabella measurements appeared slightly higher in females, these differences were not statistically significant (p> 0.05). We interpret these small discrepancies as the result of individual anatomical variation rather than true sexual dimorphism, particularly because our sample included only six female cadavers, which limits generalizability. Importantly, sesamoid bones such as the fabella do not follow the typical patterns of skeletal sexual dimorphism. Their presence, size, and ossification level are influenced much more strongly by local biomechanical loading, genetic predisposition, and tendon–muscle mechanics rather than by sex-related differences in overall bone size. As noted by Berthaume and Bull (ref. 15), the capacity to form a fabella is genetically determined, while its dimensions are shaped by environmental and mechanical stressors. Accordingly, the slightly higher measurements in females in our dataset likely reflect individual biomechanical variability combined with the small female sample size, rather than representing a biologically meaningful trend.

We have added this explanation to the Discussion section to improve clarity.

12. Table 4 last column should be prevalence

Author’s Response:

Thank you for the correction. The last column of Table 4 has been revised and is now correctly labeled as “Prevalence.” The updated version is included in the revised manuscript.

Authors’ response to Reviewer: 2

Introduction:

Reviewer's Comment: Condense the repeated statements about fabella prevalence and Asian populations.

Authors' Response:

We thank the reviewer for this suggestion. As recommended, we have condensed the repeated statements regarding fabella prevalence in Asian populations into a single, more concise sentence in the Introduction/Discussion (as appropriate) to improve the flow and avoid redundancy. However, since our other reviewer also requested the same correction, it has been highlighted in yellow in the text.

The revised text now states:

"In the literature, fabella prevalence is generally reported to be higher in Asian populations, and recent studies further support that it is particularly common in Asian countries [3, 5-7]."

Reviewer's Comment: Clarify cultural reasoning: why the higher prevalence in Asian populations is thought to be associated with habitual kneeling and squatting postures and which may increase mechanical stimulation and ossification of the fabella.”

Authors' Response:

Thank you for this valuable comment. In line with your suggestion, we have expanded upon the cultural and biomechanical rationale for the higher prevalence of the fabella in Asian populations and incorporated this explanation into the Introduction. In the revised text, we clarify that kneeling and squatting postures—common in many Asian societies—expose the fabellar region to sustained mechanical compression against the posterior aspect of the lateral femoral condyle. Furthermore, in accordance with Wolff's law, we state that such repetitive loading facilitates the ossification of the primordial cartilaginous template of the fabella. The relevant section has been highlighted in light green in the revised manuscript for your convenience.

Reviewer's Comment: Last paragraph can be ended with more clearer research aim /objectives

Authors' Response:

We thank the reviewer for this valuable comment. As suggested, the final paragraph of the Introduction has been revised to state the research aims more clearly and explicitly. The new text now reads:

"The purpose of this study was to (1) define the morphometric properties of the fabella via cadaveric dissection and (2) investigate its prevalence, lateralization pattern, gender distribution, and association with knee osteoarthritis using radiological examination in the Turkish population. This study aimed to address the lack of literature data on this subject specific to the Turkish population."

We believe this revision provides a much clearer and more direct statement of the study's objectives. We thank the reviewer for the constructive feedback.

Results

Reviewer's Comment: Provide confidence intervals (if available) for key prevalence figures

Authors' Response:

In line with the recommendation, 95% confidence intervals (CIs) have now been calculated and added for all key prevalence values reported in the manuscript. The Results section has been updated accordingly and now includes the following figures:

• Overall radiological prevalence: 36.5% (95% CI: 35.0%–38.0%)

• Female prevalence: 35.6% (95% CI: 33.9%–37.3%)

• Male prevalence: 39.5% (95% CI: 36.4%–42.6%)

• Cadaveric prevalence: 18.4% (95% CI: 9.8%–30.0%)

Additionally, as suggested, the 95% CI for the prevalence difference between the osteoarthritis and control groups has been incorporated into the Discussion section (OA – Control: 1.9%; 95% CI: –1.1% to 4.9%). Since this interval includes zero, it supports our conclusion that there is no statistically significant association between fabella presence and knee osteoarthritis.

All confidence intervals have been incorporated into the revised manuscript and the corresponding additions have been highlighted for the reviewer’s convenience.

Reviewer's Comment: Add a brief linking statement before each table to improve flow, For example-Table 1 summarizes the distribution of fabella by sex.

Authors' Response:

Thank you for your valuable feedback and for suggesting we add linking statements to improve the flow of the manuscript.

We are pleased to inform you that we have already implemented this suggestion. As per your recommendation, we have added concise introductory sentences before each table to clearly signpost their content. The added statements are as follows:

• Table 1 is now introduced with: "Table 1 summarizes the distribution of fabella by sex."

• Table 2 is now introduced with: "Table 2 presents the distribution of unilateral and bilateral fabellae, as well as right- and left-sided occurrence, in patient and healthy individuals."

• Table 3 is now introduced with: "Table 3 provides cadaver-based fabella prevalence values previously reported in the literature."

• Table 4 is now introduced with: "Table 4 summarizes radiology-based prevalence values from different populations worldwide."

We agree that these additions significantly enhance the readability and logical structure of the results section, and we thank you for this constructive comment.

Reviewer's Comment: Use table only legends when it is shown in the table (such as asterisks in table-2 used in the legend, but it is not shown in the content values)

Authors' Response:

Thank you very much for pointing out the inconsistency in the legend of Table 2. Following your feedback, we conducted a thorough re-check of our statistical tables and their corresponding legends.

You are correct. We identified that the legend for Table 2 contained a note referencing statistical symbols (such as * and **) that were not actually used in the table itself. To resolve this inconsistency, we have ensured the legend aligns precisely with the symbols used in the table by removing all explanations for symbols not present in the data presentation.

Reviewer's Comment: In table -3, use of symbol % - should be in consistent with that of text. Generally it is used after number (eg. 14%.... ) Kindly check this

Authors' Response:

Thank you for your valuable comment regarding the "consistency in the use of the % symbol." As per your suggestion, we have reviewed Table 3 and corrected all values in the "Prevalence" column to ensure they consistently follow the number-first format (e.g., 27%) throughout the manuscript.

Discussion

Reviewer's Comment: Some paragraphs repeat details from the Results. Avoid repetition to make it concise and consider to condense to focus on interpretation.

Authors' Response:

We agree that some paragraphs in the Discussion section contained unnecessary repetitions of results. As suggested, we have thoroughly revised the Discussion to eliminate these redundancies. The focus has been shifted from re-stating the findings to providing a deeper interpretation of their significance.

Reviewer's Comment: You may add a paragraph explicitly discussing possible biomechanical explanations for the lack of association between fabella and OA.

Authors' Response:

We appreciate your valuable feedback. In line with your comment, we have added a special paragraph

---

## [Editor Report · Decision Letter 1]

21 Dec 2025

Prevalence and Morphometry of the Fabella in Cadaveric and Radiological Studies: Is There a Link to Knee Osteoarthritis?

PONE-D-25-50561R1

Dear Dr. Ebru YOLACAN,

We’re pleased to inform you that your manuscript has been judged scientifically suitable for publication and will be formally accepted for publication once it meets all outstanding technical requirements.

Kind regards,

Srinivasa Rao Sirasanagandla

Academic Editor

PLOS One

---

## [Editor Report · Acceptance letter]

PONE-D-25-50561R1

PLOS One

Dear Dr. Yolaçan,

I'm pleased to inform you that your manuscript has been deemed suitable for publication in PLOS One. Congratulations! Your manuscript is now being handed over to our production team.

Kind regards,

on behalf of

Dr. Srinivasa Rao Sirasanagandla

Academic Editor

PLOS One